# How to Characterize The Landscape of Overparameterized Convolutional Neural Networks

**Yihong Gu**[*1], **Weizhong Zhang**[*2], **Cong Fang**[1], **Jason D. Lee**[1], **Tong Zhang**[2]

[1] Princeton University, [2] Hong Kong University of Science and Technology

{yihongg,jasonlee}@princeton.edu, zhangweizhongzju@gmail.com
fangcong@pku.edu.cn, tongzhang@tongzhang-ml.org

## Abstract

For many initialization schemes, parameters of two randomly initialized deep neural networks (DNNs) can be quite different, but feature distributions of the hidden nodes are similar at each layer. With the help of a new technique called *neural network grafting*, we demonstrate that even during the entire training process, feature distributions of differently initialized networks remain similar at each layer. In this paper, we present an explanation of this phenomenon. Specifically, we consider the loss landscape of an overparameterized convolutional neural network (CNN) in the continuous limit, where the numbers of channels/hidden nodes in the hidden layers go to infinity. Although the landscape of the overparameterized CNN is still non-convex with respect to the trainable parameters, we show that very surprisingly, it can be reformulated as a convex function with respect to the feature distributions in the hidden layers. Therefore by reparameterizing neural networks in terms of feature distributions, we obtain a much simpler characterization of the landscape of overparameterized CNNs. We further argue that training with respect to network parameters leads to a fixed trajectory in the feature distributions.

## 1 Introduction

A good characterization of the landscape is very important for understanding remarkable successes of DNNs in various domains including computer vision, nature language processing and speech. There have been a lot of research [29, 21, 24, 20, 26, 8] analyzing the landscape of deep neural network in parameter space. On one hand, even one-hidden-layer neural network has complex landscape with exponential number of local minima and saddle points. On the other hand, some interesting empirical phenomena like *mode connectivity*, that low-cost solutions of DNN could be connected by simple path in parameter space, have been found, indicating that the loss landscape is not as complex as we expected. These two conclusions of the landscape in parameter space seem to be contradictory.

We believe that the loss landscape should be investigated in the feature distribution space directly instead of the parameter space as existing studies have done due to the excellent capability of DNN to learn effective feature representations. But our understanding on the feature distribution learned by DNNs is still limited. For example, even to the basic question of to what extent two feature representations learned by different DNNs are essentially the same, various studies such as [23, 32, 19] reached different conclusions due to the redundancy in the feature representations comes from the special properties of DNNs, e.g., the permutation & scale invariance of the trainable parameters, the truncation operation in ReLU, etc. Two feature representations that look quite different at current layer could be essentially the same. As the example illustrated in Fig. 1(b), the difference may have little effect on the features in subsequent layers and could be alleviated, or even eliminated completely after passing the linear transformations and the activation functions ReLU

---

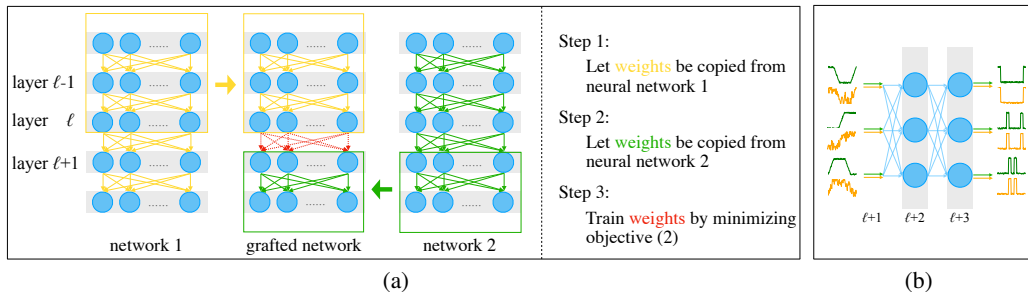

Figure 1: (a) The flowchart of neural network grafting. (b) An example of redundancy. The yellow and green curves left represent the features in layer $\ell + 1$ learned in two nets. They look different but are essential similar since they become the same after passing two more layers.

in the subsequent layers. Although several remedies are proposed in recent studies, the redundancy cannot be completely eliminated and different amounts of redundancy may lead to inconsistent conclusions. This discourages the researchers from investigating DNNs in feature distribution space.

In this paper, we start from the problem of inconsistent conclusions above and propose an effective technique named *neural network grafting* (NNG) for comparing feature distributions (see Section 4). We consider the most direct way to identify whether the feature representations learned in two hidden layers of two networks are essentially the same. The key idea is to check whether the representation learned by one network can be used in the other one without significant sacrifice in accuracy, i.e., whether one of these two networks can be grafted onto the other at that hidden layer (Fig. 1(a)). Using NNG, we find that feature distributions learned by two networks with the same architecture but different initilizations are almost the same during the *whole* training process. That means their solution paths, if we view them in the aspect of feature distributions, are the same, while they seem chaotic and quite different under the view of trainable parameters (see Section 6.1).

Our surprising finding on the uniqueness of the solution path implies that NNs become much easier to understand if we view them in the aspect of feature distributions instead of trainable parameters. This motivates us to reparameterize NNs with respect to the feature function distributions to obtain a simpler loss landscape. Specifically, we first propose a new method to reformulate a CNN as an approximation to a continuous CNN, which is obtained by letting the number of channels/hidden nodes in each hidden layer go to infinity. We then extend the emerging technique for reformulating fully connected NNs [4, 5] to continuous CNN to reparameterize it with respect to feature distributions (See Section 5). We show that although the loss of continuous CNN is still non-convex with respect to the trainable parameters, it becomes convex with respect to the feature distributions. Therefore, we obtain a much simper characterization for the loss landscape.

Although our convexity is reformulated in the feature distribution space instead of the trainable parameter space, it has significant implications on NN optimization. In fact, it can be shown that under suitable conditions, the training algorithms (e.g., SGD) of DNNs converge to a solution that is a stationary point of the convex reformulation, as shown in Section B of the Appendix. This excludes bad local minimum solutions for DNN optimization. Moreover, the convexity and unique solution path imply that CNNs are much simpler if we view them in the feature distribution space. All these demonstrate the value of studying DNNs in feature distribution space.

**Notations.** For a positive integer $n$, we let $[n]$ to be the set $\{1, 2, \ldots, n\}$. We denote $\|x\|_1$ and $\|x\|$ to be the $\ell_1$ and $\ell_2$ norm of a vector $x$ in $\mathbb{R}^d$. For any $k \in [d]$, we let $x_k$ be the value of the $k$-th dimension of $x$ and denote $vec(w)$ to be the operator for reshaping a matrix or tensor $w$ into a vector.

## 2 Related Work

### 2.1 Characterization of DNNs' Landscape

One main purpose of the studies on landscape characterization is to explain why DNNs with massive parameters can be trained efficiently in practice. They mainly explore the following subjects:

**Geometry of loss surface** The training of DNNs depends highly on the network architecture, optimization techniques (e.g., batch normalization (BN)) and some other considerations. Some researchers try to study the impacts of these factors on the geometry of the loss surface. They [8, 21] show that the loss landscapes quickly transit from being nearly convex to being highly chaotic when

the network becomes deeper, and residual connections promote surface smoothness and prevent the explosion of non-convexity, which verifies that skip connection is essential for extremely deep networks. Moreover, [28] demonstrates that the effectiveness of BN comes from its effects on the smoothness of the landscape instead of controlling *covariate shift*.

**Properties of local minima** These studies attempt to understand the phenomenon observed in practice that local minima found by training algorithms are nearly always good. They show that in idealized settings, the local minima of DNNs have similar loss values [12, 11]. To be precise, [18] proved that linear networks have no sub-optimal local minima. [30] shows that for deep residual network with a single output unit, the objective values of all local minima are no higher than what can be obtained with a linear predictor. However, these existing studies are restricted to strong assumptions of model simplifications, such as linear activation. For nonlinear DNNs in practice, theoretical properties of local minima are still unknown.

**Mode Connectivity** The phenomenon of mode connectivity implies that the optima of DNNs may have special and simple structures. Some efforts have been made to understand it theoretically. [26] shows that for overparameterized DNNs with piece-wise linear activations, the sublevel sets of their loss functions are connected and bounded. For nonlinear DNNs, [20, 27, 31] prove the existence of these low-cost paths under the assumptions such as the DNNs are dropout stable or noise stable.

Despite the above breakthroughs on characterizing the landscapes of DNNs, many empirical phenomena of DNNs are still poorly understood. Some studies [32, 19] even had inconsistent conclusions, e.g., whether randomly initialized DNNs with the same architecture learn the same representation.

## 2.2 Overparameterized Deep Learning Theory

The theoretical work about overparameterized NN, motivated by its great empirical success, can be roughly divided into two categories, i.e., mean field limit and neural tangent kernel (NTK) limit, based on whether the network parameters change a lot during training. The NTK limit [17, 10, 14, 22, 13, 2, 3, 6, 7] considers a tiny neighborhood of the initialization, and with an appropriate scaling that approaches infinity, it can be shown that the global minimum is achieved within the tiny neighborhood (called NTK regime) that becomes infinitesimal in the limit. Therefore the neural network can be linearized around the initialization, and the resulting formulation becomes convex. However, it is observed that NNs within the NTK region does not perform well compared to fully trained NNs using standard methods, which always go out of the NTK regime. Therefore NTK cannot be used to study the standard NN training process and the associated landscape characterization, as we are interested in doing in this paper. To overcome this discrepancy between theory and practice, the mean field limit [24, 9, 25] has been proposed. This is also the approach most related to this paper, because we also consider the standard training process that goes out of the NTK regime. The original mean field framework works with distributions over NN parameters, which is difficult to extend to deep networks. This paper considers the convex reformulation idea in [4, 5], where overprameterized NNs are parameterized using feature distributions of the hidden neurons. We investigate the empirical justifications and their relationship to the theory of convex reformulation.

## 3 Basics

In this section, we give a detailed formulation of standard CNN, and hereafter we refer to it as discrete CNN in contrast to our continuous CNN in Section 5.1. We notice that fully connected layer can be viewed as a convolutional layer by treating each hidden node as a special channel valued in $\mathbb{R}^{1 \times 1}$. Therefore, we can write the fully connected and convolutional layers into a unified form to simplify the formulas. Below we define a $(L+1)$-layer CNN mapping an input into a real valued vector in $\mathbb{R}^K$. The number of hidden nodes/channels in each hidden layer $\ell$ is denoted to be $m^{(\ell)}$.

We denote all the trainable parameters in the network as $\theta = (w^{(1)}, \ldots, w^{(L)}, u)$, and define the neural network recursively using them. Given an input $x \in \mathbb{R}^{W^{(0)} \times H^{(0)} \times C}$, let $m^{(0)} = C$ and define the nodes in the input later as $\hat{f}_j^{(0)}(\theta; x) = x_{:,:,j}$ with $j \in [m^{(0)}]$. For the hidden layer $\ell \in [L]$, the

output of its $j$-th hidden node/channel $\hat{f}_j^{(\ell)}(\theta;x) \in \mathbb{R}^{W^{(\ell)} \times H^{(\ell)}}$ with $j \in [m^{(\ell)}]$ could be written as

$$\hat{f}_j^{(\ell)}(\theta;x) = h^{(\ell)}\left(\hat{g}_j^{(\ell)}(\theta;x)\right) \text{ with } \hat{g}_j^{(\ell)}(\theta;x) = \frac{1}{m^{(\ell-1)}} \sum_{k=1}^{m^{(\ell-1)}} \hat{f}_k^{(\ell-1)}(\theta;x) * w_{j,k}^{(\ell)},$$

where $w_j^{(\ell)} = [w_{j,1}^{(\ell)}, \dots, w_{j,m^{(\ell-1)}}^{(\ell)}] \in \mathbb{R}^{a^{(\ell)} \times b^{(\ell)} \times m^{(\ell-1)}}$ is the $j$-th kernel, $h^{(\ell)} = \mathcal{P}^{(\ell)} \circ \psi^{(\ell)}$ with $\psi^{(\ell)}$ being the activation function and $\mathcal{P}^{(\ell)}$ being the pooling operator if layer $\ell$ is followed by a pooling layer otherwise $\mathcal{P}^{(\ell)}$ is an identity mapping. For the top layer, we denote $u_j$ to be a vector valued in $\mathbb{R}^K$, then the output $\hat{f}(\theta;x) \in \mathbb{R}^K$ can be defined as $\hat{f}(\theta;x) = \frac{1}{m^{(L)}} \sum_{j=1}^{m^{(L)}} u_j \hat{f}_j^{(L)}(\theta;x)$.

The goal of training a CNN is to minimize the following objective function:

$$\hat{Q}(w,u) = J(\hat{f}) + \hat{R}(w,u), \text{ with } \hat{R}(w,u) = \sum_{\ell=1}^{L} \lambda^{(\ell)} \hat{R}^{(\ell)}(w^{(\ell)}) + \lambda^{(u)} \hat{R}^{(u)}(u), \qquad (1)$$

where $J(\hat{f}) = \mathbb{E}_{x,y} \phi(\hat{f}(x),y)$ with $\phi(\cdot,\cdot)$ being the loss function and $\hat{R}(w,u)$ is the the regularizer with $\lambda^{(\ell)}$ and $\lambda^{(u)}$ being its non-negative hyper-parameters. In this paper, we focus on the following $\ell_{1,2}$ regularier proposed in [4] due to its ability in learning efficient features:

$$\hat{R}^{(\ell)}(w^{(\ell)}) = \frac{1}{m^{(\ell-1)}} \sum_{k=1}^{m^{(\ell-1)}} r_2 \left( \frac{1}{m^{(\ell)}} \sum_{j=1}^{m^{(\ell)}} r_1\left(w_{j,k}^{(\ell)}\right) \right), \quad \hat{R}^{(u)}(u) = \frac{1}{m^{(L)}} \sum_{j=1}^{m^{(L)}} r^{(u)}(u_j),$$

where $r_1(w) = \|vec(w)\|_1$, $r_2(w) = w^2$ and $r^{(u)}(u) = \|u\|_2^2$.

## 4 Neural Network Grafting

In this section, we introduce a new method called *neural network grafting* (NNG) to check whether two different CNNs learn similar feature representations at each layer. The intuition is that if two networks $\theta_1$ and $\theta_2$ learn similar feature representations at layer $\ell$, then the features of $\theta_1$ at layer $\ell$ can be used in $\theta_2$ without significant error increasing, that is, the first $\ell$ layers of $\theta_1$ with the trained parameters in them can be grafted onto the last $L - \ell$ layers of $\theta_2$ by only training the parameters at the joint. As shown in Fig.1(a), NNG is conducted by first copying parameters from $\theta_1$ and $\theta_2$ to construct a new network $\tilde{\theta}_{(1,2)} = (\theta_1^{(1)}, \cdots, \theta_1^{(\ell)}, \tilde{\theta}_{(1,2)}^{(\ell+1)}, \theta_2^{(\ell+2)}, \cdots, \theta_2^{(L)}, u_2)$ and then training the parameters $\tilde{\theta}_{(1,2)}^{(\ell+1)}$ to align the neurons at the joint by minimizing the following feature matching loss:

$$\min_{\tilde{\theta}_{(1,2)}^{(\ell+1)}} \frac{1}{N} \sum_{i=1}^{N} \sum_{k=1}^{m^{(\ell+1)}} \|\hat{g}_k^{(\ell+1)}(\tilde{\theta}_{(1,2)};x_i) - \hat{g}_k^{(\ell+1)}(\theta_2;x_i)\|_2^2. \qquad (2)$$

Detailed steps are given in Alg. 1 in Appendix. We could imagine that if the representation at layer $\ell$ in $\theta_1$ is similar to that of $\theta_2$, then the loss in (2) would be low after grafting, and the final validation error of $\tilde{\theta}_{(1,2)}$ will be similar to that of $\theta_2$ and otherwise it will be significantly higher than $\theta_2$.

**Remark 1.** *In our grafting process, we minimize the feature difference at layer $\ell + 1$ instead of minimize the final loss to compare the feature representations more directly.*

Notice that both our NNG and previous metrics (e.g., [23], [19] and [32]) can address the case when two feature representations are quite different but can be matched after a linear transformation. However, our NNG can address at least another important case that existing metrics cannot deal with. To be precise, if two feature representations are quite different but the difference is not important for subsequent layers, they would be regarded to be different by previous metrics while they are essentially similar. Such difference could be alleviated and even eliminated completely by, for example, the small weight and truncation of ReLU in subsequent layers. An example is shown in Fig. 1(b), where green and yellow curves represent feature representations of network $\theta_2$ and $\tilde{\theta}_{(1,2)}$ respectively. We could found that the feature representations of $\tilde{\theta}_{(1,2)}$ at layer $\ell + 1$, the first layer after grafting, is a little bit different from the features of $\theta$ at layer $\ell + 1$. However, when we take a look at the feature representations of $\tilde{\theta}_{(1,2)}$ and $\theta_2$ at layer $\ell + 2$, they are almost same. The differences are 'filtered out' by the mapping from layer $\ell + 2$ to layer $\ell + 3$. Detailed quantitative evaluation for it is presented in Fig. 2(c).

# 5 Landscape Characterization

In this section, we present our landscape characterization based on feature distributions at each layer. We first propose a general idea of continuous CNN, which could be approximated using overparameterized CNN by letting the number of channels/hidden nodes in each hidden layer go to infinity (Section 5.1). Then we reformulate it with respect to the feature distributions to obtain a simpler landscape characterization (Section 5.2).

## 5.1 Continuous CNN

Following the mean-field limit of overparameterized NN, below we give the definition of continuous CNN by letting the number of the channels/hidden nodes in each layer of a discrete CNN go to infinity. Firstly, for the input layer we denote $\mathcal{Z}^{(0)} = [C]$ to be its channel space corresponding to the $C$ channels of the input $x$. We let $\rho^{(0)}$ be a probability measure on $\mathcal{Z}^{(0)}$ and unless otherwise specified, in this paper, we fix $\rho^{(0)}$ to be a uniform distribution on $\mathcal{Z}^{(0)}$. For each $z^{(0)} \in \mathcal{Z}^{(0)}$, we let

$$f^{(0)}(\rho, z^{(0)}; x) = x_{:,:,z^{(0)}}.$$

For layer $\ell \in [L]$, let $\mathcal{Z}^{(\ell)}$ be the space of the hidden nodes/channels in layer $\ell$ equipped with probability measure $\rho^{(\ell)}$. Denote $w(z^{\ell}, z^{(\ell-1)}) \in \mathbb{R}^{a^{(\ell)} \times b^{(\ell)}}$ to be the weight/kernel connecting the nodes/channels $z^{(\ell-1)} \in \mathcal{Z}^{(\ell-1)}$ and $z^{(\ell)} \in \mathcal{Z}^{(\ell)}$, the output of hidden nodes/channels can be defined as

$$f^{(\ell)}(\rho, z^{(\ell)}; x) = h^{(\ell)}\left(g^{(\ell)}(\rho, z^{(\ell)}; x)\right),$$

where $z^{(\ell)} \in \mathcal{Z}^{(\ell)}$ and $g^{(\ell)}(\rho, z^{(\ell)}; x) = \int f^{(\ell-1)}(\rho, z^{(\ell-1)}; x) * w(z^{(\ell)}, z^{(\ell-1)}) d\rho^{(\ell-1)}(z^{(\ell-1)})$.

At last, by denoting $u(z^{(L)}) \in \mathbb{R}^K$ to be the weight connecting the node $z^{(L)}$ and the final output, then the final (fully connected) output layer is given by

$$f(\rho, u; x) = \int u(z^{(L)}) f^{(L)}(\rho, z^{(L)}; x) d\rho^{(L)}(z^{(L)}).$$

The objective function then takes form of

$$Q(\rho, u, w) = J(f) + R(\rho, u, w) \tag{3}$$

$$\text{where } R(\rho, u, w) = \sum_{\ell=1}^{L} \lambda^{(\ell)} R^{(\ell)}(w, \rho, w) + \lambda^{(u)} R^{(u)}(\rho, u, w)$$

$$\text{with } R^{(\ell)}(\rho, w) = \int r_2 \left(\int r_1\left(w(z^{(\ell)}, z^{(\ell-1)})\right) d\rho^{(\ell)}(z^{(\ell)})\right) d\rho^{(\ell-1)}(z^{(\ell-1)}),$$

$$R^{(u)}(\rho, u) = \int r^{(u)}\left(u(z^{(L)})\right) d\rho^{(L)}(z^{(L)}).$$

**Discussion:** Discrete CNN can be constructed from a continuous CNN by sampling $m^{(\ell)}$ hidden nodes/channels $\{z_1^{(\ell)}, \ldots, z_{m^{(\ell)}}^{(\ell)}\}$ for each layer $\ell$ according to the probability measure $\rho^{(\ell)}$ and letting the parameter connecting $z_i^{(\ell)}$ and $z_j^{(\ell-1)}$ be $w(z_i^{(\ell)}, z_j^{(\ell-1)})$ for all $i \in [m^{(\ell)}]$ and $j \in [m^{(\ell-1)}]$. From [4] we know that when $m^{(\ell)} \to \infty$ for all $\ell \in [L]$, the final output of discrete CNN converges to that of the continuous one. Therefore, discrete CNN is an approximation to a continuous CNN.

## 5.2 Reformulate Continuous CNNs

The uniqueness of the solution path, i.e., the unique feature distribution evolution path, discovered by our technique NNG (Sections 4 and 6.1) implies that DNNs could be much simpler if we view them with respect to feature distributions. Therefore, we reformulate continuous CNN with respect to the distribution of pre-activation of neurons, i.e., feature distribution, to eliminate the redundancy caused by weights and obtain a simpler loss landscape characterization, which is inspired from the emerging techniques [4, 5] for reformulating fully connected DNNs. First, we define

$$\mathcal{V}^{(0)} = \{v^{(0)} : v^{(0)} = [x_{:,:,i}^1, \ldots, x_{:,:,i}^N], i \in [C]\}, \text{ and } p^{(0)}(v^{(0)} = [x_{:,:,i}^1, \ldots, x_{:,:,i}^N]) = 1/C, i \in [C].$$

For $\ell \in [L]$, we denote $v^{(\ell)}$ to be the pre-activation of neuron $z^{(\ell)}$ at $N$ training samples, i.e., $v^{(\ell)} = [g^{(\ell)}(\rho, z^{(\ell)}; x_1), \ldots, g^{(\ell)}(\rho, z^{(\ell)}; x_N)] = [v_1^{(\ell)}, \cdots, v_N^{(\ell)}] \in \mathbb{R}^{W^{(\ell)} \times H^{(\ell)} \times N}$ and let $\mathcal{V}^{(\ell)}$ be the space of $v^{(\ell)}$, i.e., $\mathcal{V}^{(\ell)} = \{v^{(\ell)} : z^{(\ell)} \in \mathcal{Z}^{(\ell)}\}$. We denote $p^{(\ell)}(v^{(\ell)})$ to be the probability density function of $v^{(\ell)}$. For any $v^{(\ell-1)}$ and $v^{(\ell)}$, we let $w(v^{(\ell)}, v^{(\ell-1)}) \in \mathbb{R}^{a^{(\ell)} \times b^{(\ell)}}$ be the weight connecting them. Problem (3) can then be reformulated as

$$\min_{w,p,u} \frac{1}{N} \sum_{i=1}^{N} \phi\left(f(x_i), y_i\right) + R(w, p, u) \tag{4}$$

$s.t.\ 1)\ \sum_{v^{(0)} \in \mathcal{V}^{(0)}} v_i^{(0)} * w(v^{(1)}, v^{(0)}) p^{(1)}(v^{(1)}) p^{(0)}(v^{(0)}) = p^{(1)}(v^{(1)}) v_i^{(1)}, i \in [N];$

$2)\ \int h^{(\ell-1)}(v_i^{(\ell-1)}) * w(v^{(\ell)}, v^{(\ell-1)}) p^{(\ell)}(v^{\ell}) p^{(\ell-1)}(v^{(\ell-1)}) dv^{(\ell-1)} = v_i^{(\ell)} p^{(\ell)}(v^{(\ell)}), \ell \geq 2, i \in [N];$

$3)\ \int p^{(\ell)}(v^{(\ell)}) dv^{(\ell)} = 1 \text{ and } p^{(\ell)}(v^{(\ell)}) \geq 0,$

where $f(x_i)$ and $R(w, p, u)$ are defined as

$$f(x_i) = \int h^{(L)}(v_i^{(L)}) u(v^{(L)}) p^{(L)}(v^{(L)}) dv^{(L)}, \; R(w, p, u) = \sum_{\ell=1}^{L} \lambda^{(\ell)} R^{(\ell)}(w, p) + \lambda^{(u)} R^{(u)}(u, p),$$

with $R^{(1)}(w, p) = \sum_{v^{(0)} \in \mathcal{V}^{(0)}} r_2 \left( \int r_1 \left( w(v^{(1)}, v^{(0)}) p^{(0)}(v^{(0)}) p^{(1)}(v^{(1)}) \right) dv^{(1)} \right) / p^{(0)}(v^{(0)}),$

$R^{(\ell)}(w, p) = \int r_2 \left( \int r_1 \left( w(v^{(\ell)}, v^{(\ell-1)}) p^{(\ell-1)}(v^{(\ell-1)}) p^{(\ell)}(v^{(\ell)}) \right) dv^{(\ell)} \right) / p^{(\ell-1)}(v^{(\ell-1)}) dv^{(\ell-1)},$

$\ell \geq 2$ and $R^{(u)}(u, p) = \int r^{(u)} \left( u(v^{(L)}) p^{(L)}(v^{(L)}) \right) / p^{(L)}(v^{(L)}) dv^{(L)}.$

The weights $w$ in problem (4) are determined by $p^{(\ell)}$ via a series of constraints in 1) and 2), hence the network can be regarded as being parameterized by $p^{(\ell)}$ without $w$. Therefore, the redundancy from $w$ in networks parameterized via $w$ is eliminated. One issue with Problem (4) is that it is still non-convex due to its non-convex constraints. Fortunately, we find that by reparameterizing the above colored items according to Eqn.(5), the constraints above become either convex or linear, and then problem (4) can be rewritten into a simpler convex form. We summarized the main point in the following theorem and details are presented in Appendix A.

**Theorem 1.** *If we change the variables $u$ and $w$ as follows:*

$$\begin{cases} \text{(i) } \tilde{w}(v^{(\ell)}, v^{(\ell-1)}) = w(v^{(\ell)}, v^{(\ell-1)}) p^{(\ell)}(v^{(\ell)}) p^{(\ell-1)}(v^{(\ell-1)}), \ell \in [L]; \\ \text{(ii) } \tilde{u}(v^{(L)}) = u(v^{(L)}) p^{(L)}(v^{(L)}); \end{cases} \tag{5}$$

*then problem (4) becomes a convex optimization problem with respect to $\tilde{w}, p$ and $\tilde{u}$.*

**Discussion:** To the best of our knowledge, we give the first global convex formulation in Theorem 1 for continuous CNNs, i.e., a much simpler loss landscape characterization than existing methods. Although we find this convexity in feature distribution space instead of trainable parameter space, this would not overshadow its significant value. The reasons are as follows:

- As empirically shown in Section 6.1, the solution paths of DNNs during training, if we view them in the feature distribution space, are unique although they seem chaotic and quite different in the parameter space, which indicates that DNNs could be more understandable and simpler if we analyze them from feature distribution space.

- The unique solution path implies that the training algorithms like gradient descent (GD) could essentially work on the feature distribution space instead of the parameter space, which together with the convexity naturally explain why one usually does not observe bad local minima in practice. In fact, as shown in Section B of the appendix, under suitable conditions, gradient descent working on $(w, u)$ in the original formulation Problem 4 will converge to a solution that is a stationary point of the convex reformulation.

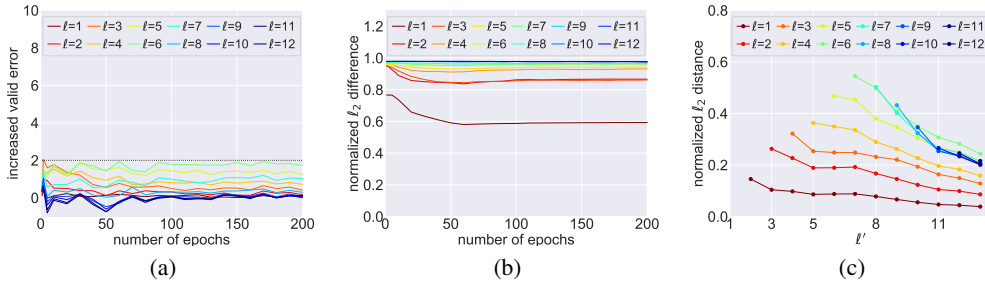

Figure 2: (a) The increased validation error of $\tilde{\theta}_{(1,2)}$ compared with $\theta_2$ when grafted at layer $\ell$ and (b) the normalized $\ell_2$ distance between the weights $\theta_1^{(\ell)}$ and $\theta_2^{(\ell)}$. These values are plotted at different training stages. (c) The mean $\ell_2$ distance (Eqn. (6)) between feature function $f^{(\ell')}(\theta_2; x)$ and $f^{(\ell')}(\tilde{\theta}_{(1,2)}; x)$ for each $\ell' > \ell$, where $x$ enumerates the training data. $\tilde{\theta}_{(1,2)}$ is grafted at layer $\ell$.

## 6 Experiments

We start from using our proposed technique NNG to show that two overparameterized CNNs with same architecture and initialization method, though have different initializations at the beginning of training, learn the unique solution path during the whole training process, which could be explained by our theory in Section 5.1. Then we give some empirical evidences for the convexity of overparameterized CNN by showing the uniqueness of its optimal solution and visualizing its the loss landscape in Section 6.2. All our empirical findings are consistent across a range of architectures and datasets, we only present the results on CIFAR-10 with VGG-16 below and postpone the results on other architectures and datasets to appendix. In appendix, we also provide results to show that VGG is overparameterized enough, i.e., a good approximation for the corresponding continuous VGG, demonstrating the applicability of our results. We use $\ell_{1,2}$ regularizer, and save intermediate checkpoints of NN parameters at time-step $t \in \{1, 2, 5, 8\} \bigcup \{10k : k \in \mathbb{N}^+\}$ in the entire section.

### 6.1 Uniqueness of Solution Path during Training

We first demonstrate that the solution paths are the same over the whole training process if we view them in terms of feature distributions, although they seem chaotic and quite different if viewed from the trainable parameters. Firstly, two VGG-16 are trained from two different initializations using the same initializer. The same initializer is used to guarantee the nets are initialized with almost the same feature distribution. For each checkpoint pair of $\theta_1$ and $\theta_2$ saved at the same time-step, we graft $\theta_1$ to $\theta_2$ at each layer and plot the increased validation error of $\tilde{\theta}_{(1,2)}$. For comparison, we also calculate the distance between the parameters in the checkpoints of $\theta_1$ and $\theta_2$. The details for calculating the distance is in Section D of appendix. The results are shown in Fig. 2.

The low increased validation errors of $\tilde{\theta}_{(1,2)}$ at all time-steps in Fig. 2(a) indicate that the feature distributions learned by $\theta_1$ and $\theta_2$ at different layers are equal to each other along the 'training trajectory'. At the same time, the normalized $\ell_2$ distances between $\theta_1$ and $\theta_2$ in Fig. 2(b) show that the RMSE between two weight matrices after alignment and unit-variance normalization is above 0.6 for all layers and time-steps, which implies that these two weight matrices are not equivalent under row/column exchange operation. A natural explanation is that the training algorithms (e.g., SGD) essentially work on feature distribution space instead of parameter space, and the evolution of the feature distributions during training follows some differential equation. Since the initializer gives almost the same initializations to the differential equation, the solution paths would be very similar during the whole process of solving this equation, i.e., the training process.

Compared with existing metrics which directly compare the feature representations at each layer, our NNG technique could partly alleviate the effect of redundancy. A direct empirical investigation is shown in Fig. 2 (c), where the mean $\ell_2$ distance between feature functions of nets $\tilde{\theta}_{(1,2)}$ and $\theta_2$ at

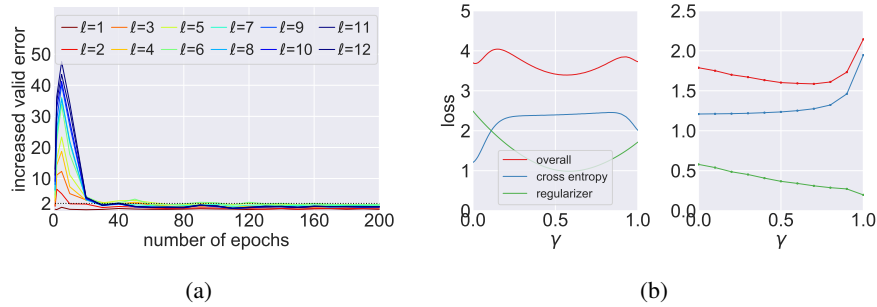

(a)                                                                                                 (b)

Figure 3: (a) The increased validation error of $\tilde{\theta}_{(1,2)}$ compared with $\theta_2$ over the training process, where $\theta_1$ and $\theta_2$ are trained with different initialization methods. (b) The overall/cross entropy/regularizer loss of $\hat{\theta}_\gamma = \gamma\theta_1 + (1-\gamma)\theta_2$ (left) and $\bar{\theta}_\gamma$ (right).

layer $\ell'$ is defined as:

$$\frac{1}{m^{(\ell')}} \sum_{k=1}^{m^{(\ell')}} \frac{1}{\sqrt{s_k}} \sqrt{\sum_{i=1}^{n} \frac{1}{n} \|\hat{f}_k^{(\ell')}(\tilde{\theta}_{(1,2)}; x_i) - \hat{f}_k^{(\ell')}(\theta_2; x_i)\|_2^2}, \tag{6}$$

and $s_k = \sum_{i=1}^{W^{(\ell')}} \sum_{j=1}^{H^{(\ell')}} \hat{\text{var}}[(\hat{f}_k^{(\ell')}(\theta_2; X))_{i,j}]$ while $\hat{\text{var}}$ stands for the estimated variance. For example, we could see that if we graft the sixth layer of $\theta_1$ to seventh layer of $\theta_2$ (line denoted as $\ell = 6$), the $\ell_2$ distance between feature functions of $\tilde{\theta}_{(1,2)}$ and that of $\theta_2$ in layer 7 is very large. But this error significantly decreases as it propagates to the output layer. Such phenomenon indicates that a large fraction of error introduced by grafting at layer $\ell$ is caused by redundancy, which could be largely cancelled out by, for example, the small weight and the truncation of activation function in each layer $\ell' > \ell$. Moreover, we also show that successful graft is non-trivial and provide more interesting empirical findings about NNG in Appendix.

### 6.2 The Evidence of Convexity

We further design some experiments to support our claim that the landscape of overparameterized CNN is convex with respect to the distribution of feature functions at each layer. To be specific, we consider two network $\theta_1$ and $\theta_2$ with different initialization methods. In this section, we use two types of variance scaling initializer [15] with uniform (*he_uniform*) and normal (*he_normal*) distribution for weights in $\theta_1$ and $\theta_2$ respectively. Because $\theta_1$ and $\theta_2$ use different initialization methods, their initial feature distribution at each layer is quite different, the training journey of $\theta_1$ and $\theta_2$ then follow two different paths. We are interested in the following two questions: (1) Is the end points of the two paths are the same? (2) From the perspective of feature distribution, does linear interpolation between two points in these two paths lead to a convex loss curve? It will strongly support our claim of convexity of landscape if the answers to these two questions are 'yes'.

**Uniqueness of Optimal Solution** To answer the first question, we make use of our NNG technique again to check whether two intermediate checkpoints of $\theta_1$ and $\theta_2$ with same time-step have similar feature distributions. To be specific, for saved checkpoints, we graft each layer $\ell$ of $\theta_1$ onto layer $\ell + 1$ of $\theta_2$ and the results are in Fig. 3(a). We can see that the increased valid error of $\tilde{\theta}_{(1,2)}$ grafted at each layer is very high in the beginning and decreases quickly as training process goes on, implying that the feature distributions learned by $\theta_1$ and $\theta_2$ start from different points and converge to a same point.

**Loss Landscape Visualization** We next consider to visualize the loss curve of linear interpolation between two points in feature distribution view. Given two NNs $\theta_1$ and $\theta_2$ with feature distributions denotes as $p_1$ and $p_2$, the intuitive idea of constructing a NN $\bar{\theta}_\gamma$, which has feature distribution $\gamma p_1 + (1-\gamma)p_2$, is to make sure the pre-activation set of $\bar{\theta}_\gamma$ is close to the mixture of that of $\theta_1$ and $\theta_2$ with ratio $\gamma/(1-\gamma)$, where the pre-activation set of $\theta$ on training set is defined as

$$\hat{\mathcal{V}}_\theta^{(\ell)} = \left\{ v_j^{(\ell)} = [\hat{g}_j^\ell(\theta; x_1), \dots, \hat{g}_j^\ell(\theta; x_N)] : j \in [m^{(\ell)}] \right\}.$$

An optimization-based algorithm is designed to construct $\bar{\theta}_\gamma$, i.e., Alg. 2 in Appendix. We let $\theta_1$ and $\theta_2$ to be VGG-16 trained after 2 epochs using *he_uniform* and *he_normal* initializers, attaining 19% and 57% accuracy respectively, and plot the loss of $\bar{\theta}_\gamma$ when $\gamma$ varies. The result is shown in Fig. 3 (b). Compared with the non-convex loss curve of linear interpolation in weight space (left), the linear interpolation in feature distribution space (right) leads to a convex loss curve.

## 7   Conclusions

In this work, we presented an experimental technique NNG, which can be used to determine whether two NNs learn similar features. Extensive experimental results with NNG suggest that overparameterized CNN learns a fixed trajectory in feature distributions. We then proposed a framework to reformulate the loss w.r.t. the feature distribution at each layer, to explain the phenomenon. This reformulation explains convex loss landscape and fixed feature distribution trajectory for SGD. Because of their better theoretical and empirical properties, we argue that NN loss landscapes should be characterized with respect to the feature distribution space rather than the parameter space.

## 8   Broader Impact

The authors feel that the broader impact seems not applicable to this work. The reason is that this is a research paper, which provides new insights to help people better understand deep neural networks and can motivate more efficient training algorithms in the future. But it is hard to say who may benefit and who may be put at disadvantage from this research.

## Acknowledgments and Disclosure of Funding

This work is supported by GRF 16201320.

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
