[Supplementary Material]

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

This appendix can be divided into four parts. To be precise,

1. Section A gives the detailed proof for Theorem 1;

2. Section B gives high-level proof of the claim that SGD will converge to the stationary point of our proposed convex formulation.

3. Section C shows omitted algorithms and calculation methods in the paper, including (1) algorithm of *neural network grafting* (2) algorithm to construct NN with mixture features of two networks.

4. Section D presents experimental configuration of this paper;

5. Section E provides more experimental evidences of the uniqueness of solution path, feature redundancy and the uniqueness of optimal solution.

6. Section F provides evidence to show that standard VGG is overparamterized enough. To be specific, we show that the feature distribution learned by standard VGG, have no difference though the *entire* training process with respect to the one learned by a larger VGG, which has same architecture except has twice number of channels/hidden notes at each layer.

7. Section G presents some additional interesting results found by our proposed NNG in comparing the feature distributions of NN at different cases, including: 1) comparing feature distributions learned in different layer $\ell_1$ and $\ell_2$; 2) comparison of feature distribution learned with two different datasets; 3) one of $\theta_1$ and $\theta_2$ is not fully trained. These results not only show that our proposed NNG metric could differentiate the case when two feature distributions for comparison is different, but also provide some very interesting empirical phenomena we find by using our NNG technique.

## A  Proof for Theorem 1

We first present the new formulation of Problem (4) after reprameterizing it according to Eqn.(5) below:

$$\min_{\tilde{w},p,\tilde{u}} \frac{1}{N} \sum_{i=1}^{N} \phi\left(f(x_i), y_i\right) + \tilde{R}(\tilde{w}, p, \tilde{u}), \tag{7}$$

$$s.t.\ 1)\ \int h^{(\ell-1)}(v_i^{(\ell-1)}) * \tilde{w}(v^{(\ell)}, v^{(\ell-1)}) dv^{(\ell-1)} = v_i^{(\ell)} p^{(\ell)}(v^{(\ell)}),\ \ 2 \leq \ell \leq L, i \in [N];$$

$$2)\ \sum_{v^{(0)} \in \mathcal{V}^{(0)}} v_i^{(0)} * \tilde{w}(v^{(1)}, v^{(0)}) = v_i^{(1)} p^{(1)}(v^{(1)}), i \in [N];$$

$$3)\ \int p^{(\ell)}(v^{(\ell)}) dv^{(\ell)} = 1 \text{ and } p^{(\ell)}(v^{(\ell)}) \geq 0,$$

where the reformualted $f(x_i)$ and $\tilde{R}(\tilde{w}, p, \tilde{u})$ take forms of

$$f(x_i) = \int h^{(L)}(v_i^{(L)}) \tilde{u}(v^{(L)}) dv^{(L)}, \ \tilde{R}(\tilde{w}, p, \tilde{u}) = \sum_{\ell=1}^{L} \lambda^{(\ell)} \tilde{R}^{(\ell)}(\tilde{w}, p) + \lambda^u \tilde{R}^{(u)}(\tilde{u}, p),$$

$$\text{with } \tilde{R}^{(1)}(\tilde{w}, p) = \sum_{v^{(0)} \in \mathcal{V}^{(0)}} r_2 \left( \int r_1 \left( \tilde{w}(v^{(1)}, v^{(0)}) \right) dv^{(1)} \right) / p^{(0)}(v^{(0)}),$$

$$\tilde{R}^{(\ell)}(\tilde{w}, p) = \int r_2 \left( \int r_1 \left( \tilde{w}(v^{(\ell)}, v^{(\ell-1)}) \right) dv^{(\ell)} \right) / p^{(\ell-1)}(v^{(\ell-1)}) dv^{(\ell-1)}, \ell \geq 2,$$

$$\tilde{R}^{(u)}(\tilde{u}, p) = \int r^{(u)} \left( \tilde{u}(v^{(L)}) \right) / p^{(L)}(v^{(L)}) dv^{(L)}.$$

To prove Theorem 1, we need to prove that problem (7) is convex. First, we need the following lemma.

**Lemma 1.** $\frac{x^2}{y}$ *is convex on* $(x, y) \in [0, +\infty) \times (0, +\infty)$.

The lemma above can be verified by the definition of convex function directly. Now we turn to give the detailed proof of Theorem 1.

*Proof.* Firstly, it is easy to verify that all the constraints 1) to 3) are convex *w.r.t.* $\tilde{w}, \tilde{u}$ and $p$.

For the objective function, we have that the loss $\phi$ is convex *w.r.t.* $\tilde{u}$. Therefore, we only need to prove the regularizer $\tilde{R}(\tilde{w}, p, \tilde{u}) = \sum_{\ell=1}^{L} \lambda^{(\ell)} \tilde{R}^{(\ell)}(\tilde{w}, p) + \lambda^u \tilde{R}^{(u)}(\tilde{u}, p)$ is convex *w.r.t.* $\tilde{w}, \tilde{u}$ and $p$. Below, we will prove each item in $\tilde{R}$ is convex. We fist denote

$$\tilde{\Psi}^{(\ell)}(\tilde{w}, p; v^{(\ell-1)}) = \int r_1 \left( \tilde{w}(v^{(\ell)}, v^{(\ell-1)}) \right) dv^{(\ell)},$$

then:

(1) For the item $\tilde{R}^{(1)}(\tilde{w}, p)$, recall that

$$\tilde{R}^{(1)}(\tilde{w}, p) = \sum_{v^{(0)} \in \mathcal{V}^{(0)}} r_2 \left( \tilde{\Psi}^{(1)}(\tilde{w}, p; v^{(0)}) \right) / p^{(0)}(v^{(0)})$$

Since we pick $r_1(w) = \|vec(w)\|_1$ in this paper, we can have that $\tilde{\Psi}^{(\ell)}$ is non-negative and convex.

Moreover, we notice that $p^{(0)}$ is constant and $r_2(w) = w^2$ in this paper, it is easy to know that $\tilde{R}^{(1)}(\tilde{w}, p)$ is convex.

(2) When $\ell \geq 2$, $\tilde{R}^{(\ell)}(\tilde{w}, p)$ takes the form of

$$\tilde{R}^{(\ell)}(\tilde{w}, p) = \int \frac{r_2 \left( \tilde{\Psi}^{(\ell)}(\tilde{w}, p; v^{(\ell-1)}) \right)}{p^{(\ell-1)}(v^{(\ell-1)})} dv^{(\ell-1)}.$$

We denote

$$I(\tilde{w}, p) = \frac{r_2 \left( \tilde{\Psi}^{(\ell)}(\tilde{w}, p; v^{(\ell-1)}) \right)}{p^{(\ell-1)}(v^{(\ell-1)})},$$

and let $p_\gamma = \gamma p_1 + (1 - \gamma) p_2$ and $\tilde{w}_\gamma = \gamma \tilde{w}_1 + (1 - \gamma) \tilde{w}_2$ for any $p_1, p_2, \tilde{w}_1, \tilde{w}_2$ and $\gamma \in [0, 1]$. Then, we have

$$
\begin{aligned}
I(\tilde{w}_\gamma, p_\gamma) &= \frac{r_2 \left( \tilde{\Psi}^{(\ell)}(\tilde{w}_\gamma, p_\gamma; v^{(\ell-1)}) \right)}{p_\gamma^{(\ell-1)}(v^{(\ell-1)})} \\
&\leq \frac{r_2 \left( \gamma \tilde{\Psi}^{(\ell)}(\tilde{w}_1, p_1; v^{(\ell-1)}) + (1 - \gamma) \tilde{\Psi}^{(\ell)}(\tilde{w}_2, p_2; v^{(\ell-1)}) \right)}{p_\gamma^{(\ell-1)}(v^{(\ell-1)})} \\
&\leq \gamma \frac{r_2 \left( \tilde{\Psi}^{(\ell)}(\tilde{w}_1, p_1; v^{(\ell-1)})) \right)}{p_1^{(\ell-1)}(v^{(\ell-1)})} + (1 - \gamma) \frac{r_2 \left( \tilde{\Psi}^{(\ell)}(\tilde{w}_2, p_2; v^{(\ell-1)})) \right)}{p_2^{(\ell-1)}(v^{(\ell-1)})} \\
&= \gamma I(\tilde{w}_1, p_1) + (1 - \gamma) I(\tilde{w}_2, p_2).
\end{aligned}
\tag{8}
$$

The first inequality in (8) holds since $\tilde{\Psi}$ is convex and the the second one comes from Lemma 1. Thus $I(\tilde{w}, p)$ is a convex function of $(\tilde{w}, p)$ and therefore $\tilde{R}^{(\ell)}(\tilde{w}, p)$ is convex.

417 (3) We denote $\tilde{u}_\gamma = \gamma\tilde{u}_1 + (1-\gamma)\tilde{u}_2$, $p_\gamma = \gamma p_1 + (1-\gamma)p_2$ for any $\tilde{u}_1, \tilde{u}_2, p_1, p_2$ and $\gamma \in [0,1]$
418 and recall that

$$\tilde{R}^{(u)}(\tilde{u}, p) = \int r^{(u)}\left(\tilde{u}(v^{(L)})\right)/p^{(L)}(v^{(L)})dv^{(L)}.$$

419 Hence, we have

$$\begin{aligned}
\tilde{R}^{(u)}(\tilde{u}_\gamma, p_\gamma) &= \int \frac{\left(\|\tilde{u}_\gamma(v^{(L)})\|\right)^2}{p_\gamma^{(L)}(v^{(L)})}dv^{(L)} \\
&\leq \int \frac{\left(\gamma\|\tilde{u}_1(v^{(L)})\| + (1-\gamma)\|\tilde{u}_2(v^{(L)})\|\right)^2}{p_\gamma^{(L)}(v^{(L)})}dv^{(L)} \\
&\leq \int \gamma\frac{\|\tilde{u}_1(v^{(L)})\|^2}{p_1^{(L)}(v^{(L)})} + (1-\gamma)\frac{\|\tilde{u}_2(v^{(L)})\|^2}{p_2^{(L)}(v^{(L)})}dv^{(L)} \\
&= \gamma\tilde{R}^{(u)}(\tilde{u}_1, p_1) + (1-\gamma)\tilde{R}^{(u)}(\tilde{u}_2, p_2).
\end{aligned}$$

420 The last inequality comes from Lemma 1 and therefore $\tilde{R}^{(u)}(\tilde{u}, p)$ is a convex function of $(\tilde{u}, p)$

421 The proof is complete. $\qquad\square$

# B  Convergence Analysis

423 We would like to study the relationship of the convex reformulation and gradient descent in the
424 original parameter space for continuous DNN. We consider the constrained formulation of continuous
425 DNN in (4) and its convex relaxation in (5), with the reparameterization of $(w, u)$ by $(\tilde{w}, \tilde{u})$.

426 In the continuous DNN formulation (4), the network is parameterized with respect to $(w, p, u)$.
427 However, the typical optimization using SGD or GD modify the network parameters $(w, u)$ but does
428 not modify $p$ directly. It can be easily seen that changing $(w, u)$ modifies $p$ accordingly.

429 When the continuous NN converges, then it reaches a point $(w_*, p_*, u_*)$ which is stationary with
430 respect to the neural network parameters. That is, if we modify $(w_*, u_*)$, the objective value does
431 not decrease (however, we are not allowed to modify the distribution $p_*$ directly because it is not an
432 explicit part of the DNN parameters, but rather it is an hidden parameter induced by the constraints).

433 Nevertheless the following result shows that under suitable conditions, this point is also stationary
434 with respect to the distributional parameter $p_*$. That is, if we vary $(w_*, p_*, u_*)$ arbitrarily, as long as
435 the new parameters satisfy the constraints, then the objective value does not decrease.

436 Since the same claim holds for the reparameterization $(\tilde{w}_*, p_*, \tilde{u}_*)$ in (5). This result shows that
437 when the continuous DNN converges, it converges to a solution of the convex reformulation, and this
438 establishes a connection between the standard DNN optimization method using SGD, and our convex
439 reformulation.

440 **Theorem 2.** *Consider an arbitrary point $(w_*, p_*, u_*)$ of the constrained formulation of continuous*
441 *NN in (4), such that varying the DNN network parameters $(w_*, u_*)$ cannot decrease the objective*
442 *function. Assume that $p_*^{(\ell)}(v) \neq 0$ for all $v$ and $\ell$, then $(\tilde{w}_*, p_*, \tilde{u}_*)$ is a stationary point of the convex*
443 *reformulation reparameterized by (5).*

444 *Proof.* (High-level Sketch) We present a highlevel proof that is easier to understand. A more rigorous
445 treatment requires heavy notations in differential equations and functional analysis, which we shall
446 avoid in this version.

447 The key argument is to show that all varations of $(\tilde{w}_*, p_*, \tilde{u}_*)$ that satisfy the constraints can be
448 achieved by varying $(w_*, u_*)$ of the continuous DNN parameters, which is commonly used in gradient
449 descent optimization algorithms.

450 Let the objective function be $\tilde{Q}(\tilde{w}, p, \tilde{u}) = Q(w, p, u)$.

451 To simplify notations, we denote the DNN weights on the $\ell$-th layer $w(v^{(\ell)}, v^{(\ell-1)})$, and the reparam-
452 eterized weight $\tilde{w}(v^{(\ell)}, v^{(\ell-1)})$ by $w^{(\ell)}$ and $\tilde{w}^{(\ell)}$.

453 Note that we have
$$\langle \nabla_{\tilde{u}} \tilde{Q}(\tilde{w}_*, p_*, \tilde{u}_*), \Delta \tilde{u} \rangle = 0$$

454 for all $\Delta \tilde{u}$, since $Q(w_*, p_*, u_*)$ is a stationary point with respect to network parameter $u$ and $p_* > 0$.
455 Here the inner product is with respect to integration over $v^{(L)}$.

456 Since the constraints at layer $\ell$ involve the weight $\tilde{w}^{(\ell)}$ and density $p^{(\ell)}$, therefore to prove the desired
457 result, we only need to show that for each $\ell$:

$$\langle \nabla_{\tilde{w}^\ell} \tilde{Q}(\tilde{w}_*, p_*, \tilde{u}_*), \Delta \tilde{w}_*^{(\ell)} \rangle + \langle \nabla_{p^{(\ell)}} \tilde{Q}(\tilde{w}_*, p_*, \tilde{u}_*), \Delta p_*^{(\ell)} \rangle = 0, \tag{9}$$

458 under the infinitesimal transformation (with other layers fixed):

$$\tilde{w}_*^{(\ell)} \to \tilde{w}_*^{(\ell)} + \epsilon \Delta \tilde{w}_*^{(\ell)}, \qquad p_* \to p_*^{(\ell)} + \epsilon \Delta p_*^{(\ell)},$$

459 when $\epsilon \to 0$, with the constraint

$$\int h^{(\ell-1)}(v_i^{(\ell-1)}) \Delta \tilde{w}_*^{(\ell)}(v^{(\ell)}, v^{(\ell-1)}) \, dv^{(\ell)} = v^{(\ell)} \Delta p_*^{(\ell)}(v^{(\ell)}).$$

460 Here inner product with respect to $\Delta \tilde{w}_*^{(\ell)}$ is integration with respect to $(v^{(\ell)}, v^{(\ell-1)})$, and inner
461 product with respect to $\Delta p_*^{(\ell)}$ is is integration with respect to $v^{(\ell)}$.

462 Since the constraint is linear, we can decompose the subspace into two subspaces. Case 1 is
463 $\Delta p_*^{(\ell)}(v^{(\ell)}) = 0$, and Case 2 is $\Delta p_*^{(\ell)}(v^{(\ell)}) \neq 0$ but $\int \Delta p_*^{(\ell)}(v^{(\ell)}) dv^{(\ell)} = 0$ and

$$\Delta \tilde{w}_*^{(\ell)}(v^{(\ell)}, v^{(\ell-1)}) = \tilde{w}_*(v^{(\ell)}, v^{(\ell-1)}) \Delta p_*^{(\ell)}(v^{(\ell)}) / p_*^{(\ell)}(v^{(\ell)}). \tag{10}$$

464 It is easy to check that the constraints are satisfied in Case 2.

465 Now to show (9), we only need to show it under these two subspaces, as detailed below.

466 **Case 1**: In this case, we can let

$$\Delta w_*^{(\ell)}(v^{(\ell)}, v^{(\ell-1)}) = \Delta \tilde{w}_*^{(\ell)}(v^{(\ell)}, v^{(\ell-1)}) \frac{1}{p^{\ell-1}(v^{(\ell-1)}) p^{(\ell)}(v^{(\ell)})},$$

467 and consider a change of network parameter

$$w_*^{(\ell)} \to w_*^{(\ell)} + \Delta w_*^{(\ell)}.$$

468 This change of parameter does not change $u_*$ and $p_*$ in the resulting neural network. Since
469 $Q(w_*, p_*, u_*)$ is stationary in $w_*$, it follows that

$$\langle \nabla_{w_*^{(\ell)}} Q(w_*, p_*, u_*), \Delta w_*^{(\ell)} \rangle = 0,$$

470 which implies (9).

471 **Case 2**: In this case, we consider a change of DNN parameter $w^{(\ell)}$, which leads to

$$w_*^{(\ell)}(v^{(\ell)}, v^{(\ell-1)}) \to w_+^{(\ell)}(v_+^{(\ell)}, v^{(\ell-1)}) = w_*^{(\ell)}(v_+^{(\ell)}, v^{(\ell-1)}) + \epsilon g_\ell(v^{(\ell)}) \omega(v^{(\ell-1)}),$$

472 where variable $v_+$ is induced by

$$v_+^{(\ell)} = v^{(\ell)} + \epsilon \cdot g_\ell(v^{(\ell)}),$$

473 where we simply pick an arbitrary vector function $\omega(v)$ so that

$$\int h^{(\ell-1)}(v_i^{(\ell-1)}) \omega_i(v^{(\ell-1)}) p^{(\ell-1)}(v^{(\ell-1)}) dv^{(\ell-1)} = 1.$$

474 With the above transformation, we have the constraint:

$$\int h^{(\ell-1)}(v_i^{(\ell-1)}) * w_+(v_+^{(\ell)}, v^{(\ell-1)}) p_+^{(\ell)}(v_+^{(\ell)}) p^{(\ell-1)}(v^{(\ell-1)}) dv^{(\ell-1)} = p_+^{(\ell)}(v_+^{(\ell)}) [v_+^{(\ell)}]_i, \ell \geq 2, i \in [N].$$

The corresponding change of distribution is from $p^{(\ell)}(v)$, to $p^{(\ell)}_+(v_+)$, which is equivalent to the distribution of $p^{(\ell)}(v)$ under the change of variable $v \to v_+$, corresponding to the DNN resulted from a change of network parameter $w^{(\ell)}_* \to w^{(\ell)}_+$.

Using the Fokker-Planck equation, we have

$$p^{(\ell)}_+(v) = p^{(\ell)}_*(v) - \epsilon \nabla(p^{(\ell)}_*(v)g_\ell(v)) + o(\epsilon),$$

and let

$$\Delta p^{(\ell)}_*(v) = p^{(\ell)}_+(v) - p^{(\ell)}_*(v) = -\epsilon \nabla(p^{(\ell)}_*(v)g_\ell(v)) + o(\epsilon).$$

It follows that we can keep $\tilde{w}^{(\ell+1)}$ unchanged by setting

$$w^{(\ell+1)}_*(v^{(\ell+1)}, v^{(\ell)}) \to w^{(\ell+1)}_*(v^{(\ell+1)}, v^{(\ell)})\frac{p^{(\ell)}(v^{(\ell)})}{p^{(\ell)}_+(v^{(\ell)})}.$$

Note also that

$$\int h^{(\ell-1)}(v^{(\ell-1)}_i)*w_*(v^{(\ell)}_+, v^{(\ell-1)})p^{(\ell)}_+(v^{(\ell)}_+)p^{(\ell-1)}(v^{(\ell-1)})dv^{(\ell-1)} = p^{(\ell)}_+(v^{(\ell)}_+)[v^{(\ell)}_+]_i, \ell \geq 2, i \in [N].$$

Therefore

$$\int h^{(\ell-1)}(v^{(\ell-1)}_i) * \Delta \tilde{w}'_*(v^{(\ell)}_+, v^{(\ell-1)}) = 0,$$

where

$$\Delta \tilde{w}'_*(v^{(\ell)}_+, v^{(\ell-1)}) = [w_+(v^{(\ell)}_+, v^{(\ell-1)}) - w_*(v^{(\ell)}_+, v^{(\ell-1)})]p^{(\ell)}_+(v^{(\ell)}_+)p^{(\ell-1)}(v^{(\ell-1)}).$$

Now case 1 implies that

$$\langle \nabla_{\tilde{w}}\tilde{Q}(\tilde{w}_*, p_*, \tilde{u}_*), \Delta \tilde{w}'_* \rangle = 0. \tag{11}$$

Note that the stationarity of the DNN network with respect to a change of the network parameter $w^{(\ell)}_*$ implies that as $\epsilon \to 0$:

$$\langle \nabla_w Q(w_*, p_*, u_*), \Delta w_* \rangle + \langle \nabla_{p^{(\ell)}} Q(w_*, p_*, u_*), \Delta p^{(\ell)}_* \rangle = o(\epsilon),$$

and $\Delta w_* = w_+ - w_*$. This equation in the convex reformulation is equivalent to

$$\langle \nabla_{\tilde{w}^{(\ell)}}\tilde{Q}(\tilde{w}_*, p_*, \tilde{u}_*), [\tilde{w}^{(\ell)}_+ - \tilde{w}^{(\ell)}_*] \rangle + \langle \nabla_{p^{(\ell)}} Q(\tilde{w}_*, p_*, \tilde{u}_*), \Delta p^{(\ell)}_* \rangle = 0.$$

By substracting (11), we obtain

$$\langle \nabla_{\tilde{w}^{(\ell)}}\tilde{Q}(\tilde{w}_*, p_*, \tilde{u}_*), \Delta \tilde{w}^{(\ell)}_* \rangle + \langle \nabla_{p^{(\ell)}} Q(\tilde{w}_*, p_*, \tilde{u}_*), \Delta p^{(\ell)}_* \rangle = 0,$$

where $\Delta \tilde{w}_*$ is given by (10). This implies (9). It follows by letting $\epsilon \to 0$ that for Case 2, (9) holds for all

$$\Delta p^{(\ell)}_*(v) = -\nabla(p^{(\ell)}_*(v)g_\ell(v)).$$

Since $g_\ell(v)$ is arbitrary, it is easy to verify that this implies that (9) holds for Case 2 with all possible $\Delta p^{(\ell)}_*(v)$ such that

$$\int \Delta p^{(\ell)}_*(v)dv = 0.$$

This proves the desired result. $\qquad\qquad\square$

# C  Missing Algorithms in Main Text

## C.1  Neural Network Graft

Here we present the detailed steps of our technique *neural network grafting* (NNG) in Algorithm 1. It takes two neural network $\theta_1$ and $\theta_2$ as input and output a grafted neural network $\theta_{(1,2)}$. In order to keep the notations simple to demonstrate our idea, we assume $\theta_1$ and $\theta_2$ have the same number of layers and the graft algorithm is to compare the feature distributions of layer $\ell$ in $\theta_1$ and $\theta_2$, i.e. graft layer $\ell$ in $\theta_1$ onto layer $\ell + 1$ in $\theta_2$. It should be emphasized that Algorithm 1 can be applied to the case where $\theta_1$ and $\theta_2$ have different numbers of hidden nodes, i.e., $m^{(\ell)}_1 \neq m^{(\ell)}_2$ in each layer, which is used in Section F, and the case where $\theta_1$ and $\theta_2$ have different numbers of hidden layers. Moreover, NNG can also even be used to compare different layers, for example, compare layer $\ell_1$ in $\theta_1$ and layer $\ell_2$ in $\theta_2$ by grafting layer $\ell_1$ in $\theta_1$ onto layer $\ell_2 + 1$ in $\theta_2$. We don't explicitly write down the algorithm of those cases and the details can be found in Section D (comparing layer with different hidden nodes) and Section G.1 (comparing different layers).

**Algorithm 1** Neural Network Graft at layer $\ell$

---

**Input:** Dataset $\{(x_i, y_i)\}_{i=1}^{N}$ and two networks $\theta_1$ and $\theta_2$ with $\theta_i = (\theta_i^{(1)}, \ldots, \theta_i^{(L)}; u_i), i = 1, 2$.

Allocate a grafted network $\tilde{\theta}_{(1,2)}$, whose front $\ell$ layers' hidden size is same with that of front $\ell$ layers in $\theta_1$ (i.e. $m_{(1,2)}^{(\ell')} = m_1^{(\ell')}$ for $\ell' \in [\ell]$), and back $L - \ell$ hidden layers' size is same with that of back $L - \ell$ layers in $\theta_2$ (i.e. $m_{(1,2)}^{(\ell')} = m_2^{(\ell')}$ for $\ell + 1 \le \ell' \le L$).

Copy weights from $\theta_1$ and $\theta_2$ to $\tilde{\theta}_{(1,2)}$, i.e., $\tilde{\theta}_{(1,2)} = (\theta_1^{(1)}, \cdots, \theta_1^{(\ell)}, \tilde{\theta}_{(1,2)}^{(\ell+1)}, \theta_2^{(\ell+2)}, \cdots, \theta_2^{(L)}, u_2)$.

Train $\tilde{\theta}_{(1,2)}^{(\ell+1)}$ by solving the following problem

$$\min_{\tilde{\theta}_{(1,2)}^{(\ell+1)}} \sum_{i=1}^{N} \sum_{k=1}^{m^{(\ell+1)}} \|\hat{g}_k^{(\ell+1)}(\tilde{\theta}_{(1,2)}; x_i) - \hat{g}_k^{(\ell+1)}(\theta_2; x_i)\|_2^2. \tag{12}$$

**Return:** The grafted network $\tilde{\theta}_{(1,2)}$ for evaluation

---

## C.2  Feature Distribution Fusion

Here we present Algorithm 2, the detailed optimization-based algorithm to construct a new neural network $\bar{\theta}_\gamma$ from two nets $\theta_1$ and $\theta_2$. The feature distribution of $\bar{\theta}_\gamma$ can be approximately equal to $\gamma p_1 + (1 - \gamma)p_2$, where $p_1$ and $p_2$ are the feature distributions of $\theta_1$ and $\theta_2$, respectively.

---

**Algorithm 2** Feature Distribution Fusion

---

**Input:** Training set $\{(x_i, y_i)\}_{i=1}^{N}$ and the sets $\hat{\mathcal{V}}_i^{(\ell)}$ of two CNNs with $\ell \in [L]$ and $i = 1, 2$.
**for** $\ell = 1, \ldots, L$ **do**

Construct a set $\hat{\mathcal{V}}_\gamma^{(\ell)}$ by sampling $\gamma m^{(\ell)}$ and $(1-\gamma)m^{(\ell)}$ points from $\hat{\mathcal{V}}_1^{(\ell)}$ and $\hat{\mathcal{V}}_2^{(\ell)}$, respectively.

Train $\bar{w}_\gamma^{(\ell)}$ in the $\ell$-th layer of VGG $\bar{\theta}_\gamma$ by minimizing:

$$L(\bar{w}_\gamma^{(\ell)}) = \frac{1}{m^{(\ell)}N} \sum_{i=1}^{N} \sum_{v_j^{(\ell)} \in \hat{\mathcal{V}}_\gamma^{(\ell)}} \|\hat{g}_j^{(\ell)}(\bar{\theta}_\gamma; x_i) - v_{j,i}^{(\ell)}\|_2^2 + \lambda^{(\ell)}\hat{R}^{(\ell)}(\bar{w}_\gamma^{(\ell)}). \tag{13}$$

**end for**
Train the output layer by solving:

$$\min_{\bar{u}} \frac{1}{N} \sum_{i=1}^{N} \phi(\hat{f}(\bar{\theta}_\gamma; x_i), y_i) + \lambda^{(u)}\hat{R}^u(\bar{u}).$$

**Return:** The trained network $\bar{\theta}_\gamma$.

---

# D  Experimental Configuration

In this section, we present the experimental settings and implementation details. We use same regularization parameter $\lambda$ for all layers, i.e., $\lambda^{(1)} = \cdots = \lambda^{(L)} = \lambda^{(u)} = \lambda$.

**Training $\theta_1$ and $\theta_2$** We denote $\theta_1$ and $\theta_2$ to be two standard VGG-16 without batch normalization, since batch normalization can change the structure of the networks. The hidden sizes of the last two fully-connected layers are both 4096. All models are implemented using Tensorflow [1] and we follow the same data augmentation as [16]. Models are trained using stochastic gradient descent with momentum 0.9. We fix all of the hyperparameters summarized in Table 1 and only tune the initial learning rate and the regularization parameter. Since training neural network with large regularization parameter is difficult to converge using the standard initial learning rate 0.01, we vary

it in $\{1e-2, 5e-3, 2e-3, 1e-3\}$ and select the largest one that could produce robust training results (actually it could attain lowest validation error for that).

Table 1: Hyper-parameters used for training VGG-16

| Hyper-parameter | Value |
|---|---|
| Batch size | 64 |
| Epochs to anneal lr | 60 |
| Anneal rate | 0.2 |
| Warm-up epochs | 1 |

**Training $\tilde{\theta}_{(1,2)}$ grafted at adjacent layers** To compare the feature functions learned at layer $\ell$ of $\theta_1$ and $\theta_2$, we need to train a grafted network $\tilde{\theta}_{(1,2)}$ connecting two adjacent layers, i.e., layer $\ell$ of $\theta_1$ and layer $\ell+1$ of $\theta_2$. The detailed training process is as follows: we first train two networks $\theta_1$ and $\theta_2$ using previous configurations and then we train the weight matrix and bias between grafted layer $\ell$ and $\ell+1$ of the network $\tilde{\theta}_{(1,2)}$ using our proposed Algorithm 1 and copy all other parameters from $\theta_1$ and $\theta_2$. We use Adam Optimizer with initial learning rate $1e-3$ to optimize the objective Eqn (2). The data augmentation is same with as that in training $\theta_1$ and $\theta_2$ and number of epochs for training is 10 for CIFAR-10 and 20 for CIFAR-100.

**Training $\tilde{\theta}_{(1,2)}$ grafted at non-adjacent layers** To determine whether feature function distributions of $\theta_1$ and $\theta_2$ ($\theta_1$ and $\theta_2$ have the same structure) at two different layers are similar or not, we need to train a network grafted at non-adjacent layer. The training process is similar to that of grafting at adjacent layers. We provide detailed implementations here. Suppose we wish to calculate the similarity of feature distributions at layer $\ell_1$ and $\ell_2$ in two nets $\theta_1$ and $\theta_2$ respectively (that is to determine whether the last $L - \ell_2$ layers in $\theta_2$ could be grafted on the first $\ell_1$ layers in $\theta_1$), we minimize the feature matching loss w.r.t. parameters $w_{(1,2)}$:

$$\frac{1}{N}\sum_{i=1}^{N}\|\mathcal{P}^o \circ \hat{f}^{(\ell_1)}(\theta_1; x_i) * w_{(1,2)} - \hat{f}^{(\ell_2+1)}(\theta_2; x_i)\|_2^2,$$

where $N$ is the number of training samples, $o$ is the number of max-pooling operator in layers $\ell' \in [\ell_1 + 1, \ell_2]$, $\hat{f}^{(\ell)}(\theta; x)$ and $\hat{g}^{(\ell)}(\theta; x)$ are pre-activation and post-activation feature map in $\mathbb{R}^{W^{(\ell)} \times H^{(\ell)} \times m^{(\ell)}}$.

**Calculating the distance between the parameters of $\theta_1$ and $\theta_2$** We first assume that the alignment permutation $q^{(\ell)}$, whose $j$-th component $q_j^{(\ell)}$ represent that neuron $j$ in layer $\ell$ of network $\theta_1$ is aligned to neuron $q_j^{(\ell)} \in [m^{(\ell)}]$ in layer $\ell$ of network $\theta_2$, is defined at each layer. We then could use the following statistic to measure the difference of $\theta_1$ and $\theta_2$ in the parameter matrix space at layer $\ell$:

$$\sum_{i=1}^{m^{(\ell)}} \sum_{j=1}^{m^{(\ell-1)}} \|(w_2^{(\ell)})_{q_i^{(\ell)}, q_j^{(\ell-1)}} - (w_1^{(\ell)})_{i,j}\|_2^2 / \left( (\|w_1^{(\ell)}\|_2^2 + \|w_2^{(\ell)}\|_2^2)/2 \right), \tag{14}$$

where $\|x\|_2$ is the $\ell_2$ norm of tensor $x$'s flattened vector, $w_1^{(\ell)}, w_2^{(\ell)}$ are weights at layer $\ell$, $(w_k^{(\ell)})_{i,j}$ is a scalar for fully-connected layers and is a flattened 9-dim vector for VGG-16. Here we consider a heuristics algorithm to determine $q^{(\ell)}$ from the input layer to the last hidden layer. For the input layer $\ell = 0$, the permutation is an identity map, i.e., $q_j^{(0)} = j$, using the nature of image. Now if the alignment permutation at layer $\ell - 1$, i.e., $q^{(\ell-1)}$, is determined, the alignment permutation at layer $\ell$ could be calculated by minimize the objective

$$\min_{q^{(\ell)}} \sum_{i=1}^{m^{(\ell)}} \sum_{j=1}^{m^{(\ell-1)}} \|(w_2^{(\ell)})_{q_i^{(\ell)}, q_j^{(\ell-1)}} - (w_1^{(\ell)})_{i,j}\|_2^2, \tag{15}$$

it should be noted that such optimization problem could be written as be a standard problem of minimum weighted bipartite matching, i.e., minimizing the following objective

$$\min_{q^{(\ell)}} \sum_{i=1}^{m^{(\ell)}} d_{j, q_j^{(\ell)}} \tag{16}$$

Figure 4: The increased validation error of $\tilde{\theta}_{(1,2)}$ compared with $\theta_2$ over the training process. The configuration of $\theta_1$ and $\theta_2$ for each subplot could be found in Table 2

Figure 5: The mean $\ell_2$ distance (Eqn. (6)) between feature function $f^{(\ell')}(\theta_2; x)$ and $f^{(\ell')}(\tilde{\theta}_{(1,2)}; x)$ for each $\ell' > \ell$, where $x$ enumerates the training data. $\tilde{\theta}_{(1,2)}$ is grafted at layer $\ell$. The configuration of $\theta_1$ and $\theta_2$ for each subplot could be found in Table 2

Figure 6: The increased validation error of $\tilde{\theta}_{(1,2)}$ compared with $\theta_2$ over the training process. $\theta_1$ and $\theta_2$ are (a) VGG-16 trained on CIFAR-100 and (b) VGG-19 trained on CIFAR-10.

, where $d_{i,j} = \sum_{k=1}^{m^{(\ell-1)}} \|(w_2^{(\ell)})_{i,q_k^{(\ell-1)}} - (w_1^{(\ell)})_{j,k}\|_2^2$. This optimization problem could be solved in $O((m^{(\ell)})^3)$ using Hungarian Algorithm.

# E    Additional Experiments on Other Datasets, Architectures, and Regularizer

In this section, we show that our empirical findings found by our NNG technique is consistent across different datasets, architectures and regularizers, and provide more explanations. We consider three type of configurations, which is listed in the following Table 2.

Table 2: Configurations for Additional Experiments

| Index | Dataset | Architecture | Regularizer |
|---|---|---|---|
| a | CIFAR-100 | VGG-16 | $\ell_{1,2}$ |
| b | CIFAR-10 | VGG-19 | $\ell_{1,2}$ |
| c | CIFAR-100 | VGG-16 | $\ell_2$ |

Figure 7: The increased validation error of $\tilde{\theta}_{(1,2)}$ compared with $\theta_2$ over the training process, where (a) $\theta_1$ is standard VGG-16 and $\theta_2$ has twice number of hidden nodes/channels compared with $\theta_1$ (b) $\theta_2$ is standard VGG-16 and $\theta_1$ has twice number of hidden nodes/channels compared with $\theta_2$. Both of the networks are trained using $\ell_{1,2}$ regularizer on CIFAR-10 dataset.

In the following, we demonstrate that our findings on 1) uniqueness of solution path; 2) the evidence of feature redundancy; 3) evidence of convexity found by NNG are consistent across these configurations. The training settings for $\theta_1$, $\theta_2$, the intermediate saved checkpoints, and training settings for $\tilde{\theta}_{(1,2)}$ are the same with respect to that in main text.

**Uniqueness of Solution Path**. We first show that the feature distributions learned by $\theta_1$ and $\theta_2$ trained from two different initializations are similar though the entire training process. Fig.4 plots the increased validation error of $\tilde{\theta}_{1,2}$ compared with $\theta_2$ as training goes on. It is observed that the error is bouned in the low-error region, indicating that $\theta_1$ and $\theta_2$ learn similar feature distribution during the entire training process in all of these configurations.

**Evidence of Feature Redundancy**. In the main text, we claim that a large amount of error, the $\ell_2$ difference between features of $\tilde{\theta}_{1,2}$ and that of $\theta_2$ in layer $\ell + 1$ introduced by grafting at layer $\ell$ is caused by redundancy, which have little effect on the feature function in subsequent layers. We show such phenomenon could also be found in the configurations above. As illustrated in Fig.5, the mean $\ell_2$ distance between $\hat{f}^{(\ell')}(\tilde{\theta}_{(1,2)}; x)$ and $\hat{f}^{(\ell')}(\theta_2; x)$ decreases as layer $\ell'$ increase.

**Convergence to Same Point** Similar to the settings in the main text, we use our NNG technique to check whether the feature distributions learned in two nets $\theta_1$ and $\theta_2$ intilizied with two different methods (i.e., $he_{uniform}$ and $he_{normal}$) converge to the same distribution as the training process goes on. The configurations of $\theta_1$ and $\theta_2$ are chosen from (a) and (b) in Table 2, i.e., $\theta_1$ and $\theta_2$ are VGG-16 trained on CIFAR-100, or $\theta_1$ and $\theta_2$ are VGG-19 trained on CIFAR0-10. For saved checkpoints, we graft each layer $\ell$ of $\theta_1$ onto layer $\ell + 1$ of $\theta_2$ and the results are presented in Fig.6. We can see that the increased valid error of $\tilde{\theta}_{(1,2)}$ grafted at each layer is very high in the beginning (since $\theta_1$ and $\theta_2$ are initialized with two quite different feature distributions) and decreases quickly as training process goes on, and finally almost all the curves finally bounded in low-error region. This implies that the feature distributions learned by $\theta_1$ and $\theta_2$ start from different points and converge to a same point.

## F  VGG is Over-parameterized Enough in Feature Distribution View

In this section, we conduct experiments to show that VGG-16 are overparameterized enough. To be precise, we verify this point by showing that the feature representations learned by VGG-16 are stable, in a sense that, the learned feature representations would be quite similar if we increase the numbers of channels/hidden nodes in each layer of VGG-16 by 2 times.

We first train two networks $\theta_1$ and $\theta_2$ on CIFAR-10, where one of $\theta_1$ and $\theta_2$ is a standard VGG-16 and the other is VGG-16 with twice number of hidden nodes/channels compared with the standard one. We compare the feature distributions learned in them during the training process by grafting the checkpoints of $\theta_1$ and $\theta_2$ saved at the same time-step together, just as we did in the experiments for showing the uniqueness of same solution path in the main text. The results are shown in Fig.7. We

could see that the increased validation of $\tilde{\theta}_{(1,2)}$ compared with $\theta_2$ is accumulated in low-error region for all layers and time-steps, which verifies our claim empirically.

# G Additional Findings

## G.1 Feature Distributions Comparison between Different Layers

We next use our proposed NNG metric to compare the similarity of feature function distributions between different layers, i.e., layer $\ell_1$ of $\theta_1$ and layer $\ell_2$ of $\theta_2$. Here $\theta_1$ and $\theta_2$ are neural network trained from two different initializations with same $\ell_{1,2}$ regularizer on same dataset, $\tilde{\theta}_{(1,2)}$ is the network grafted at two non-adjacent layers. To specific, we use the feature representations at layer $\ell_1$ in $\theta_1$ to be the input for the feature representations at layer $\ell_2 + 1$ in $\theta_2$ and train the weights and bias parameters using a $\ell_2$ regression loss, see Section D for details. The increased validation error of $\tilde{\theta}_{(1,2)}$ compared with $\theta_2$ is reported in Fig.8. From Fig. 8a), we can see that for the fully trained VGG-16 on CIFAR10, the feature representations in the layers near the input are relatively different from each other, while those learned in the layers near the output are similar with each other. We can also observe that the feature representations learned in the six layers near the input are quite different from those learned in the subsequent layers, since when two of them are grafted together, the validation error increases significantly by at least $20\%$. To the best of our knowledge, we are the first to find this property of DNNs in feature learning.

Another observation is the *measured similarity between different layers using our NNG metric is different for small and large regularization parameters*. Since the results and core conclusions are similar for CIFAR-10 and CIFAR-100, we just provide analysis for CIFAR-10 below. To be precise, Fig. 8b) shows that for large regularization parameter $\lambda = 10$, the back seven layers near output are almost the same, while when it comes to small regularization parameter $\lambda = 1$ only five layers near the output are similar.

(a) CIFAR-10, $\lambda = 1$      (b) CIFAR-10, $\lambda = 10$

(c) CIFAR-100, $\lambda = 1$      (d) CIFAR-100, $\lambda = 4$

Figure 8: The increased validation error of $\tilde{\theta}_{(1,2)}$ compared with $\theta_2$, where $\theta_1$ and $\theta_2$ are trained VGG-16 using same $\ell_{1,2}$ regularizer parameter but different initialization. For each curve denote as $\ell_1$, we plots the increased validation error when we directly graft the first $\ell_1$ layers in $\theta_1$ on the $\ell_2 + 1$-th layer in $\theta_2$, which demonstrate whether the feature functions at layer $\ell_1$ in $\theta_1$ is similar to that at layer $\ell_2$ in $\theta_2$. The shadowed regions represent the 95% confidence interval.

(a) $\lambda = 1$     (b) $\lambda = 2$     (c) $\lambda = 4$     (d) $\lambda = 8$     (e) $\lambda = 10$

Figure 9: The increased validation error of $\tilde{\theta}_{(1,2)}$ compared with $\theta_2$, where $\theta_1$ and $\theta_2$ are trained VGG-16 using same $\ell_{1,2}$ regularizer parameter but different initialization. For each curve denote as $\ell_1$ (we omit the legend of $\ell_1$ here and it is same to that in Fig 8), we plots the increased validation error when we directly graft the first $\ell_1$ layers in $\theta_1$ on the $\ell_2 + 1$-th layer in $\theta_2$, which demonstrate whether the feature functions at layer $\ell_1$ in $\theta_1$ is similar to that at layer $\ell_2$ in $\theta_2$.

618 The above observation indicates that: 1) our proposed NNG metric can differentiate the similarity
619 and difference effectively; 2) larger regularization parameter will promote more layers near the final
620 output of a neural network to learn similar feature representations.

621 To better demonstrate 2), we further analyze the results about how these curves change when $\lambda$ varies
622 from 1 to 10, the results are shown in Fig 9. We could draw two observations of curves' changes
623 if we choose $5\%$ as the threshold to differentiate similar or not. One is that the front several layers
624 are similar with their adjacent layers when $\lambda$ is small, when $\lambda$ becomes larger, these adjacent layers
625 become different. The other is more layers near the output become similar when $\lambda$ becomes larger.

626 **G.2 Feature Distributions Learned on Different Datasets**

Figure 10: The increased validation error of $\tilde{\theta}_{(1,2)}$ compared with $\theta_2$, where $\theta_1$ and $\theta_2$ are trained VGG-16 $\ell_{1,2}$ regularizer with parameter $\lambda = 1$ but different initialization and different dataset. $\theta_1$ and $\theta_2$ are trained with CIFAR-10 and CIFAR-100 respectively for curve 'c10-c100' and are trained with CIFAR-100 and CIFAR-10 respectively for the curve 'c100-c10'

627 In the section, we investigate the similarity of feature representations learned on different datasets. To
628 be specific, we train two networks $\theta_1$ and $\theta_2$ on CIFAR-10 and CIFAR-100, respectively and compare
629 the feature representations learned by them using our NNG technique. The increased validation error
630 of $\tilde{\theta}_{(1,2)}$ compared with $\theta_2$ is shown in Fig.10, from which we can draw the following conclusions:

    1. Our proposed NNG metric can clearly identify whether two feature learned from different
631         dataset are similar or not. We can see that the feature representations learned from different
632         datasets are quite similar in the layers near the input layer while have a large gap in the
633         layers near the final output.
634

2. We could see that the performance of grafted network $\tilde{\theta}_{(1,2)}$ comprised of the near-input layers of the network trained CIFAR-10 and the near-output layers of the network on CIFAR-100 is significant worse than the one comprised the near-input layers of the network on CIFAR-100 and the near-output layers on CIFAR-10, this could be contributed to two factors: 1) CIFAR-100 task is more difficult than CIFAR-10 task; 2) the feature representation learned on CIFAR-100 are more diverse.

(a) CIAFR-10, $\theta_1$ not fully trained

(b) CIAFR-10, $\theta_2$ not fully trained

(c) CIAFR-100, $\theta_1$ not fully trained

(d) CIAFR-100, $\theta_2$ not fully trained

Figure 11: Increased validation performance of $\tilde{\theta}_{(1,2)}$ compared with $\theta_2$. The shadowed regions represent the 95% confidence interval. For each subplot, $e = [u]$ represents $\theta_k$ is the model after $u$ training epochs and the other network is the converged model. The dotted line is the validation error of corresponding $\theta_k$ after $u$ training epochs.

### G.3 Feature Distributions Learned at Different Stages

In this section, we consider the case where one of $\theta_1$ and $\theta_2$ is not fully trained. Fig 11 shows that: 1) our proposed metric is identifiable to the case when one of $\theta_1$ and $\theta_2$ has not trained to optimal; 2) the performance gap between the grafted NN and the original NN quickly decreases to zero for the previous few layers and slowly decreases for the last few layers. This indicates that the feature functions in the previous few layers converge much faster than that of the last few layers.