[Reviews · NeurIPS 2020]

Review 1

Summary and Contributions: This paper studies the loss landscape of overparameterized convolutional neural networks, and proposes a new way to understand the training dynamics via the trajectories of feature distributions. The authors argue that for a fixed architecture, the feature distributions of differently initialized networks remain similar during the training process. To compare feature distributions, a new technique called network grafting is introduced, and extensive experiments show strong support for this argument.

Strengths: The perspective of understanding the training dynamics via feature distributions seems novel and appealing to me. The point that the feature distributions follow a fixed trajectory seems intriguing and the technique of network grafting looks useful to me.

Weaknesses: I wonder if the closeness of feature distributions could be measured using standard statistics such as mean and variance. See the additional feedback part.

Correctness: The claims, proofs, and experiments all look correct to me.

Clarity: The paper is well-written and easy to follow. I enjoy reading the paper very much.

Relation to Prior Work: The comparison to prior works is clearly discussed.

Reproducibility: Yes

Additional Feedback: As the paper tries to argue that the feature distributions are similar for differently initialized networks, I wonder if this can be shown by using standard statistics such as mean and variance. That is, for a fixed layer, we measure the statistics of the values of the hidden nodes, and see if different initializations give similar numbers. In fact, it may also make sense to make the comparison to networks with different number of hidden nodes at that layer. If the measures were not close, I am not sure if it would be appropriate to consider such finite networks as approximation of a common continuous CNN, with their hidden nodes sampled i.i.d. from the distribution underlying the continuous CNN. ----- I've read the rebuttal, and I am happy with the answers from the authors.


Review 2

Summary and Contributions: This paper presents an empirical way how to compare the features extracted from the same layer of a network but learned under different initializations. By doing some experiments on VGG-16, they observed that at each layer the features are similar in some sense during the entire training process. To justify this phenomenon, they consider an infinitely wide neural network and then show that the training problem (known to be non-convex in the space of parameters) can be reformulated as a convex program with respect to the distribution of features at each layer. Based on that, they argue that the landscape of overparameterized CNN is convex wrt the distribution of features at each layer.

Strengths: Understanding the landscape of convolutional neural networks is clearly an important task in the studies of deep learning, and hence the relevance of this paper. The paper contains both empirical and theoretical results which support their claims.

Weaknesses: In the paper (e.g. abstract) the authors claim that one can reformulate the problem as a convex program (as shown in section 5), and comment that this is a surprising result. However I do not find this particular result surprising. The way the current proof works is just to do a change of variable. Of course you can always define a new variable to abstract away or hide the non-convexity of the problem, so that the new problem looks like 'convex' with respect to the new variables. However, the underlying non-convexity of the problem is still there as your new variables (\tilde{w}..) become more complex. For this reason, I don't think that this result can be used as a strong theoretical support for the claims that the landscape of overparameterized CNN is convex with respect to the feature distribution at each layer, nor does it explain why one usually does not observe bad local minima in practice (c.f. L212).

Correctness: It is good that the paper has some theoretical result to support their claims. Unfortunately I think that the current theoretical support is not strong enough.

Clarity: The paper is clearly written.

Relation to Prior Work: The related work section is fine. However the authors are encouraged to add more detailed discussion along the works of [3,4,21,30,17] since they appear to be more closely related to the contribution of the present paper.

Reproducibility: Yes

Additional Feedback: Overall, I have the impression that the paper tries to advocate for the fact that the landscape of convolutional neural networks is convex with respect to the features at each layer. Unless I miss some important details in the paper, I think this claim is not true. Even for finite networks, understanding the convexity of the problem with respect to the features at each layer would require some first understanding of the space of features realizable by the network at every intermediate layer, which is also not treated here. In the main paper, the authors present a theoretical result to support their claim. However, if I understand correct the way their proof works is just by a change of variable where the new variables hide the non-convexity of the system. Thus I believe this cannot be used as a valid argument to say that the whole system is convex with respect to the original distribution of interest. My current score is given based on the current state of the paper. However I am happy to adapt it after the rebuttal if the authors can provide more convincing argument or additional results that convince me about the significance of their work. ==================== After rebuttal: I thank the authors for their response. I have raised my score a bit as I think some of the discussion here can be potentially interesting for future research. For the sake of the reader, it would be good if the authors can add some comments about what the results imply and do not imply. Some comment about finite networks may be also nice to have.


Review 3

Summary and Contributions: In this paper, the authors firstly develop a new technique named NNG for comparing the features learned in different neural networks. They then show that that feature distributions learned by two networks with the same architecture but different initializations are almost the same during the whole training process. They finally prove that the loss landscape of overparameterized CNNs after parameterizing them with respect to the feature distribution in each layer is convex. All these results send out an important signal that DNNs could be much simpler than we expected if we study them in the feature distribution space.

Strengths: The main strengths of this paper are listed below: 1) This paper presents a simpler loss landscape for CNN. 2) An interesting empirical finding: DNNs with different initializations have the same solution path in the feature distribution space during training. 3) 1) and 2) imply that DNNs could be much simpler than we expected if we study them in the feature distribution space instead of the parameter space. This would be important for the future research on understanding DNNs. 4) A novel technique named NNG for comparing the features learned in neural networks.

Weaknesses: 3. Weaknesses 1) There are a lot of notations in this paper. I recommend the authors to give a table or index in appendix for all the notations, which would make this paper more readable. 2) The empirical finding about the unique solution path is very interesting. However, more details about the experimental configuration should be provided. For example, whether the training samples/mini batches are shuffled or not in each training process. And can we have the same conclusion when the training samples are shuffled and not shuffled? 3) Some typos, e.g., ‘reformualted’ in line 396->’reformulated’. ----------------------------------------------- Reviewer has read the author response. I would like to stick to the original score.

Correctness: All the claims and the method are correct.

Clarity: This paper is well written.

Relation to Prior Work: The related work is discussed clearly.

Reproducibility: Yes

Additional Feedback: See my comments about the weaknesses.

[Author Response · NeurIPS 2020]



Figure 1: The statistics of (a) mean and (b) variance of features in layer $\ell \in [12]$ during training.

We thank all the reviewers for their constructive comments. We appreciate their suggestions like add a table of notations
(Reviewer #3) and provide more detailed discussion (Reviewer #2). We will take these suggestions in later version.

**To Reviewer #1**:
*Q1. As the paper tries to argue that the feature distributions are similar for differently initialized networks, I wonder if*
*this can be shown by using standard statistics such as mean and variance.*

A1. Yes, we agree with you that when two finite networks are sampled from a common continuous CNN, their standard
statistics would be similar. We report the difference of mean/variance of the feature functions in two different initialized
networks $\theta_1$ and $\theta_2$ in Fig 1. We notice that the nets $\theta_1$ and $\theta_2$ at initialization are exactly sampled from a common
continuous VGG-16 as we use the same initialization method, so the differences of mean/variance at initialization
can be used as a baseline, i.e., we can compare the differences during training with this baseline to see whether the
mean/variance are close. We adopt widely used statistics in hypothesis test as metrics to measure such difference. To be
specific, we use $|\bar{X} - \bar{Y}|/\sqrt{2S_p^2/m}$ for mean (smaller means closer) and $\frac{1}{m-1}\sum_{i=1}^{m}(X_i - \bar{X})^2/\frac{1}{m-1}\sum_{i=1}^{m}(Y_i - \bar{Y})^2$
for variance (closer to 1 means closer), where $S_p^2 = (\sum_{i=1}^{m}(X_i - \bar{X})^2 + \sum_{i=1}^{m}(Y_i - \bar{Y})^2)/(2m-2)$, $\{X_i\}_{i=1}^{m}$ and
$\{Y_i\}_{i=1}^{m}$ are the $m$ feature values for fixed input image in one layer of nets $\theta_1$ and $\theta_2$, respectively. We can see that the
differences of both mean and variance during training are always at the same level with those at initialization (the parts
where x-axis<0). Therefore, the results above indicate that mean and variance of the features in $\theta_1$ and $\theta_2$ are similar
during the whole training process.

Moreover, we'd like to point out that the standard statistics mean and variance are not enough to differentiate two
feature distributions. Besides the difficulty in choosing a proper threshold, there exist some networks, whose feature
functions have close mean and variance but are essentially different.

**To Reviewer #2**:
*Q1. The proposed method just does a change of variable where the new variables hide the non-convexity of the system.*

A1. The reviewer might have missed some results in our appendix. Our convex formulation obtained by a change of
variable is highly nontrivial. Below, we explain it from three aspects:

• The technique of reformulating two layer networks with respect to the distributions of hidden nodes was
proposed in mean field theory based studies, such as [8] and [22], which have been widely accepted. This
work is a nontrivial extension of this technique in multi layer neural networks.

• In appendix B, we explicitly present the connection between original formulation and convex reformulation in
Theorem 2. It claims that for arbitrary point $(w_*, p_*, u_*)$ in original formulation, if varying $(w_*, u_*)$ cannot
decrease the objective then the corresponding point $(\tilde{w}_*, p_*, \tilde{u}_*)$ after our proposed change of variable is a
stationary point in the convex reformulation. This suggests that when CNN converges (GD only updates
$(w_*, u_*)$), it converges to a stationary point of our convex reformulation. Therefore, our formulation can
explain why one usually does not observe bad local minima when the widths are in the limit of infinity.

• We do obtain a convex reformulation for continuous CNN by changing the variables $(w, u)$ in the original
formulation into $(\tilde{w}, p, \tilde{u})$, but it should be noted that the whole system is essentially determined by $(p, \tilde{u})$
instead of $(\tilde{w}, p, \tilde{u})$ since given $(p, \tilde{u})$, $\tilde{w}$ could be calculated by minimizing the regularizer $\tilde{R}(\tilde{w}, p, \tilde{u})$ under
the first two constraints in Eqn 7. Therefore, we do not hide any non-convexity in the new variable $\tilde{w}$.

**To Reviewer #3**:
*Q1. More details about training process (whether to shuffle the training data) and whether we can obtain consistent*
*results when the training data is shuffled and not shuffled in training.*

A1. We use independent shuffling and data augmentation scheme in training each network in order to obtain the same
test accuracy as previous work. To be specific, in training each network, we use the data augmentation used in previous
work for training VGG nets (we mentioned in appendix D), which means that for each epoch, the training data was
shuffled first and then standard data augmentation scheme is used (to be specific, the images are zero-padded with 4
pixels on each side, randomly cropped to produce $32 \times 32$ images, and horizontally mirrored with probability 0.5).

[Meta-Review · NeurIPS 2020]

This paper studies the landscape of overparametrized convolutional networks and argues that their training dynamics can be analyzed by comparing the trajectories of feature distributions. Using “network grafting” as a metric, it shows feature distribution trajectories of two networks with the same architecture but different initializations remain close during training. The paper also shows that although the landscape is non-convex with respect to the trainable parameters, it can be reformulated as a convex function with respect to the features. Reviewers rate the paper as top 50%, marginally above, and marginally above. They find that the paper is well written and proposes a novel and appealing perspective for analyzing training dynamics. However, there was lack of clarity about the claim of convexity, which the authors clarified in the rebuttal. I think adding those clarifications to the paper is needed. I also note there are earlier papers on the analysis of the landscape which show that the non-convex objective on the parameters can be tightly connected to a convex objective on output space (https://arxiv.org/pdf/1506.07540). Finally, I also think it would be good to add the discussion in the rebuttal about using other standard statistic metrics. Overall, there is agreement this is a good paper.